# Single-Cell Transcriptome Analysis of Radiation Pneumonitis Mice

**DOI:** 10.3390/antiox11081457

**Published:** 2022-07-26

**Authors:** Miaomiao Yang, Qiang Fan, Tom K. Hei, Guodong Chen, Wei Cao, Gang Meng, Wei Han

**Affiliations:** 1Anhui Province Key Laboratory of Medical Physics and Technology/Institute of Health and Medical Technology, Hefei Institutes of Physical Sciences, Chinese Academy of Sciences, Hefei 230031, China; yangmiaomiaom@163.com (M.Y.); qiangfan@mail.ustc.edu.cn (Q.F.); c3024034@163.com (G.C.); caowei2018bss@163.com (W.C.); 2Science Island Branch of Graduate School, University of Science and Technology of China, Hefei 230026, China; 3Clinical Pathology Center, The Fourth Affiliated Hospital of Anhui Medical University, Hefei 230022, China; menggangbl@163.com; 4Center for Radiological Research, College of Physician and Surgeons, Columbia University Medical Center, New York, NY 10032, USA; tkh1@cumc.columbia.edu; 5Collaborative Innovation Center of Radiation Medicine of Jiangsu Higher Education Institutions and School for Radiological and Interdisciplinary Sciences (RAD-X), Soochow University, Suzhou 215123, China

**Keywords:** radiation pneumonitis (RP), single-cell RNA-seq (scRNA-seq), mouse, oxidative stress

## Abstract

Radiation-induced lung injury (RILI), especially radiation pneumonitis (RP), is a common clinical complication associated with thoracic radiotherapy for malignant tumors. However, the specific contributions of each cell subtype to this process are unknown. Here, we provide the single-cell pathology landscape of the RP in a mouse model by unbiased single-cell RNA-seq (scRNA-seq). We found a decline of type 2 alveolar cells in the RP lung tissue, with an expansion of macrophages, especially the Fabp4low and Spp1high subgroup, while Fabp4high macrophages were almost depleted. We observed an elevated expression of multiple mitochondrial genes in the RP group, indicating a type 2 alveolar cell (AT2) response to oxidative stress. We also calculated the enrichment of a cGAS-STING signaling pathway, which may be involved in regulating inflammatory responses and cancer progression in AT2 cells of PR mice. We delineate markers and transcriptional states, identify a type 2 alveolar cell, and uncover fundamental determinants of lung fibrosis and inflammatory response in RP lung tissue of mice.

## 1. Introduction

Radiotherapy is the primary and adjuvant therapy for several malignancies of the thoracic cavity, despite that the lungs are very sensitive to ionizing radiation. In fact, within three years after the discovery of X-rays, the first incidence of radiation pneumonitis (RP) was reported during the treatment of tuberculosis patients in 1898 [1]. Since then, for patients undergoing thoracic radiotherapy for malignant tumors of the lung, esophagus, breast, and thymus, radiation-induced lung injury (RILI) has become a common complication [2]. The sensitivity of normal lung parenchyma to ionizing radiation is always a dose-limiting factor for radiotherapy of thoracic malignancies. With the advance in modern treatment planning and adherence to the accepted dosimetry predictors of RILI, the risk of RP has been greatly reduced, but it is still higher than 15% with a mortality rate up to 4% [3,4].

RILI can be generally divided into three stages: early/latent, acute RP, and chronic radiation pulmonary fibrosis (RPF). RP is observed between 8–16 weeks of exposure, while RPF often occurs after 1 year of exposure [5]. Although studies have shown that RP or RPF is neither completely independent pathological changes, nor events with simple linear relationships, they may start with pathological changes at the same time after irradiation. There is evidence that the typical RILI could transform into fibrosis after RP. After lung injury is caused by pneumonia, the alveolar regions experience transient abnormally elevated mechanical tension due to the loss of alveoli, which further induces alveolar stem cells to initiate a differentiation program of new alveoli formation. Mechanical tension returns to physiological levels with the formation of new alveoli. In addition, elevated mechanical tension further activates the TGF-β signaling loop in alveolar stem cells, leading to an abnormal increase of interstitial cells around alveolar stem cells and fibrotic lesions [6]. As such, it is very important to control the pathological changes as early as possible in the inflammatory stage, as RPF is irreversible once the disease process starts. 

At present, the process of RILI is known to be dynamic and complex, which could be regulated by oxidative stress, hypoxia, immunocytes, and multiple cytokines [7]. The involved cell types include type 2 alveolar cells (AT2), endothelial cells, alveolar macrophages, fibroblasts, and T helper cells. Similarly to drug-induced lung injury, radiation exposure induces the degeneration of alveolar cells and endothelial cells (EC), which includes increased capillary permeability, injuries of type 1 alveolar cell (AT1) and type 2 alveolar cell (AT2), decreased alveolar surfactant and serum protein infiltration into the alveoli. These changes are invisible to light microscopy or clinical imaging [8,9]. However, electron microscopy revealed the signs of degeneration of alveolar cells, increased mucus secretion of goblet cells, swelling of the basement membrane, and changes in endothelial cells in RILI samples [8,9]. The damaged alveolar cells secrete cytokines, recruit inflammatory cells such as macrophages and lymphocytes to the alveoli and lung interstitium, and induce acute pneumonia, indicating that the development of RP involves an interaction network among multiple types of cells within the microenvironment. However, the roles and molecular mechanisms of these various cells that mediate the inflammation effect are not completely understood, in part because the previous studies mostly relied on the analysis of bulk tissues. At the cellular level, analysis of the kinetically evolving heterogeneity inherent to disease is limited by the range of flow cytometric markers used to study different cell populations in bulk.

Recent advances in single-cell RNA sequencing (scRNA-seq) provide powerful tools for exploring the genetic and functional heterogeneity of different cell types, reconstructing evolutionary lineages, and detecting rare subpopulations [10,11]. This technique has been broadly adopted to decipher the molecular mechanisms of complex diseases such as cancers, arthritis, lupus nephritis, asthma, pulmonary fibrosis, etc. [10,11]. Not surprisingly, scRNA-seq was also employed to interpret lung diseases or injuries in recent years. Xie et al. identified the subtypes of fibroblasts in the process of pulmonary fibrosis [12]. Strunz et al. found the presence of a krt8+ transitional stem cell state in human lung fibrosis [13]. Aran et al. analyzed the heterogeneity of macrophages in drug-induced lung fibrosis and found that CX3CR1+ siglecF+ transitional macrophages were localized to the fibrotic niche and exhibited a profibrotic effect [14]. However, a study on RILI at single-cell resolution is still lacking so far. 

To explore the mechanism of RILI in an unbiased way, we performed a single-cell transcriptome analysis of the entire lung tissue and peripheral blood mononuclear cell (PBMC) from a mouse model and identified 15 cell types, many of which exhibited compositional and/or transcriptomic changes in the RP group. Our results showed a decline of type 2 alveolar cells, especially bronchioalveolar stem cells, and a slight decrease of T cells in the RP lung tissue, with an expansion of macrophages. Further analysis investigation identified distinct functional macrophage subpopulations in control and RP lung tissue, showing a transformation of macrophage function in response to irradiation. Cell–cell interaction analysis also suggested more complex interactions in the RP model. Overall, our analysis provides an overview of the subtypes of the stromal and immune cells after irradiation, and thus, elucidates the mechanisms of RILI.

## 2. Materials and Methods

### 2.1. RP Mouse Model

Six- to eight-week-old C57BL/6J mice were obtained from GemPharmatech Company (Jiangsu, China). All animal studies were conducted according to protocols approved by the Ethical Committee on the Use and Care of Experimental Animals of Hefei Institutes of Physical Science, Chinese Academy of Sciences. For irradiation, the anesthetized mice were immobilized. After the accurate positioning of the irradiation area of mice with the simulator, a single dose of 20.0 Gy of 6.0 MV X-rays was delivered to a 2.0 cm × 2.0 cm area in the whole lung at a dose rate of 2.0 Gy/min. All other parts of the animal were shielded with a custom-made lead cover. The control mice (CN) were subjected to the same treatment except for irradiation [15].

### 2.2. Quantitatively Analyze Lung Injury

Eight weeks after radiation, the mice were euthanized by hemorrhagic shock. The left lung was excised and immersed in 4% formaldehyde for at least 24 h before being embedded in paraffin. Sections (5 μm) were cut and stained with hematoxylin/eosin (H&E) according to the standard method and observed with optical microscopy. The Ashcroft scale was used for the quantitative histological analysis of pneumonia induced by irradiation. The severity of pneumonia was scored double-blinded with optical microscopy based on Acute Lung Injury in Animals Study Group [16], which scored from 0 to 8: 0, normal; 1–2, alveolar walls slightly broadening, and less than 10 inflammatory cells in high power field of vision; 3–4, alveolar walls significantly broadening, 10–20 inflammatory cells under high power field of vision; 5–6, alveolar walls significantly broadening, inflammatory cells infiltration full of alveolar walls; 7–8, in addition to inflammatory cell infiltration full of alveolar walls, obvious alveolar infiltration. Each sample was scored using nine randomly selected fields and the average score was calculated.

### 2.3. Single-Cell Preparation

Two RP mice and one healthy control mouse were prepared, and lung tissues and PBMC were collected to perform the single-cell RNA-seq (scRNA-seq). The collected lung tissues were stored in the sCelLiveTM Tissue Preservation Solution (Singleron Biotechnologies, Nanjing, China), and fresh peripheral blood from RP and healthy control mice was collected with EDTA anticoagulant tubes and then transported to the Singleron lab on ice as soon as possible after animals were euthanized. The specimens were dissociated into single-cell suspension with a Singleron PythoNTM Automated Tissue Dissociator (Singleron Biotechnologies) with sCelLiveTM Tissue Dissociation Mix (Singleron Biotechnologies), based on the preset protocol for mouse lung tissue dissociation in the dissociator. Phosphate buffered saline (PBS; Solarbio, Beijing, China) was used to dilute the whole blood sample at a ratio of 1:1, and then the sample was added into a tube with approximately 2/3 volume of Ficoll (GE Healthcare, Shanghai, China). After centrifugation at 400× *g* for 35 min, three layers were obtained based on the size and density. The middle cell suspension layer was transferred into a new 15 mL centrifuge tube, PBS was added, and then centrifuged at 300× *g* for 7 min. The supernatant was discarded, and the pellet containing PBMCs was washed twice and then resuspended in PBS. The sample was stained with trypan blue (Sigma, Shanghai, China) and microscopically evaluated for cell viability.

### 2.4. Single-Cell RNA Sequencing (scRNA-seq)

Single-cell suspensions at 1 × 10^5^ cells/mL in concentration in PBS (HyClone, Shanghai, China) were prepared and loaded onto microfluidic devices and scRNA-seq libraries were constructed according to Singleron GEXSCOPE^®^ protocol by GEXSCOPE^®^ Single-Cell RNA Library Kit (Singleron Biotechnologies) and Singleron Matrix^®^ Automated single-cell processing system (Singleron Biotechnologies). Individual libraries were diluted to 4 ng/μL and pooled for sequencing. Pools were sequenced on Illumina HiSeq X (Illumina, San Diego, CA, USA) with 150 bp paired end reads.

### 2.5. Primary Analysis, Quality control, Dimensionality Reduction, and Clustering of scRNA-seq Data

Raw reads were processed to generate gene expression profiles using a customized pipeline. Briefly, raw reads were first processed with fastQC v0.11.4 (https://www.bioinformatics.babraham.ac.uk/projects/fastqc/ accessed on 20 July 2022) and fastp to remove low-quality reads, and with cutadapt to trim poly-A tail and adapter sequences [17]. Cell barcodes and UMI were extracted. After that, we used STAR v2.5.3a to map reads to the reference genome GRCm38 (mm10). UMI counts and gene counts of each cell were acquired with feature Counts v1.6.2 software and used to generate expression matrix files for subsequent analysis. 

Before analyses, cells were filtered by UMI counts below 30,000 and gene counts between 200 and 5000, followed by removing the cells with over 20% mitochondrial content. After filtering, the functions from Seurat v3.1.2 [18] were used for dimension reduction and clustering. Then we used NormalizeData() and ScaleData() functions to normalize and scale all gene expressions, and selected the top 2000 variable genes with the FindVariableFeautres() function for PCA analysis. Using the top 20 principal components, we separated cells into multiple clusters with FindClusters(). For sub-clustering of a clustered cell type, this resolution was set at 1.2. Finally, the t-SNE algorithm or UMAP algorithm was employed to perform cells with a two-dimensional space.

### 2.6. Differentially Expressed Genes (DEGs) Analysis

To identify DEGs, we used the Seurat FindMarkers() function based on Wilcox likelihood-ratio test with default parameters [19]. Furthermore, the DEGs with each cluster combined with the expression of canonical markers were employed for the cell type annotation, and the DEGs among RP and CN for each cluster or cell type were employed for interpreting the different cellular responses in RP. 

### 2.7. Cell Type Annotation

Identification of cell type for clusters was performed according to the expression of canonical markers in the DEGs and the reference database SynEcoSys^TM^ (Singelron Biotechnologies), which contains collections of canonical cell type markers for single-cell seq data, from CellMakerDB, PanglaoDB, and recently published literature. Doublet cells were identified as expressing markers for different cell types and removed manually. Heatmaps, dot plots, and violin plots displaying the expression of markers used to identify each cell type were generated by Seurat v3.1.2 DoHeatmap()/DotPlot()/Vlnplot().

### 2.8. Pathway Enrichment Analyses

To find biological functions or pathways that were significantly associated with the genes specifically expressed, we performed the Gene Ontology (GO) [19] and Kyoto Encyclopedia of Genes and Genomes (KEGG) analysis with the “clusterProfiler” R package version [20]. Pathways with a *p*-value less than 0.05 were considered significantly enriched. 

To investigate the potential functions that are significantly associated with the specifically expressed genes, we performed GSVA pathway enrichment analysis with the hallmark gene sets from the Molecular Signatures Database, with the average gene expression of each cell type used as input data [19,20]. 

In order to perform the GSEA analysis, we ranked the DEGs of each targeted cluster by the adjusted *p*-value, which was then used as inputs to the fgsea function in the fgsea R package. We used the hallmark gene sets of the MSigDB collection as reference sets [21]. 

### 2.9. Trajectory Analysis

To understand the reprogramming processes of cells, we performed the Monocle2 method to construct the single-cell trajectories [22]. We used DEGs identified by Seurat v3.1.2 to sort cells in the spatial-temporal differentiation order. We used FindVairableFeatures() to select highly variable genes from clusters and DDRTree() for dimension-reduction. Finally, we visualized the trajectory with plot cell trajectory(). 

### 2.10. RNA Velocity Analysis

For the RNA velocity analysis, we used velocyto and scVelo in python with default parameters to annotate unspliced and spliced reads in BAM files and to estimate steady-state gene-specific velocity [23,24]. After that, we projected the data to the t-SNE plot from Seurat clustering analysis to visualize the results.

### 2.11. Cell–Cell Interaction Analysis

The cell–cell interaction analysis was performed by CellPhoneDB based on receptor-ligand interactions between two cell types/subtypes [25]. Cluster labels of all cells were randomly permuted 1000 times to calculate the null distribution of average ligand–receptor expression levels of the interacting clusters. Individual ligand or receptor expression was threshold with a cutoff value based on the average log gene expression distribution for all genes across all the cell types. The significant cell–cell interactions were defined as *p*-value < 0.05 and average log expression > 0.1, which was visualized with the circlize (0.4.10) R package [26].

### 2.12. Gene Regulatory Network Inference

To analyze transcription factor regulatory networks, we performed SCENIC R toolkit using scRNA expression matrix and transcription factors in AnimalTFDB [27]. Regulatory networks were predicted by the GENIE3 package based on the co-expression of regulators and targets. We used the RcisTarget package to search for transcription factor binding motifs in the data. Genes involved in a predicted regulatory network were defined as a gene set, whose AUC value was calculated by the AUCell package to assess the activity of the regulatory network in cells.

## 3. Results

### 3.1. Single-Cell Analysis Reveals Heterogeneity in RP

To characterize the transcriptome response to irradiation, we performed scRNA-seq on unsorted cells from the right lung and peripheral blood mononuclear cell (PBMC) of mice at 8 weeks post-radiation, respectively (Figure 1a). This time point has previously been shown to coincide with the inflammation of radiation injury [9,28]. Compared to the control group, H&E staining of the lung parenchyma showed pulmonary capillary congestion, slightly-widened alveolar septum, presence of edema, and inflammatory cells gathered in the alveolar septum, secondary bronchi, and perivascular area in the RP group (Figure 1b). These histological changes were consistent with the pathology score of pneumonia (Figure 1c).

Approximately 43,062 sequenced cells that met quality control metrics were further analyzed [29,30,31], with an average of 7177 cells obtained from each sample. There are 31 cell clusters identified by unsupervised clustering, including epithelial cells (Epcam^+^CD45^−^, Cluster 5, 13, 17), immune cells (Epcam^−^CD45^+^, cluster 0~10, 11, 14), and endothelial or stromal (Epcam^−^CD45^−^, cluster 12, 15, 16) (Appendix A). Further annotation of the cell types using SynEcoSys ^TM^ database revealed distinct lymphocytes (B cells, T cells, NK cells), myeloid cells (macrophages, monocytes, granulocyte), type 2 alveolar cell (AT2), fibroblasts, smooth muscle cells, and vascular endothelial cells (VECs) (Figure 2a,b). Marker genes used as well as the number and proportion of each identified cell type in different samples were listed in Appendix A [32,33,34]. In PBMC samples, T cells and B cells accounted for a large proportion. In contrast, more myeloid cells (macrophages, and dendritic cells) and a small amount of plasma cells were detected in tissue samples. Epithelial cells and some stromal cells only present in tissue samples [35,36]. Compared to the normal tissue samples, the infiltration of T cells in the two pneumonia tissue samples was slightly lower (17.11% in RP samples vs. 24.18% in the control sample), while the infiltration of myeloid cells was higher (40.66% in RP samples vs. 27.19% in the control sample) (Figure 2c,d). 

### 3.2. scRNA-seq Identifies Seven Lung Epithelial Cell Types and Reveals an Altered Epithelial Cell Composition in RP

Among all the analyzed cells, 1811 epithelial cells were identified (1511 AT2 and 300 club cells), and we found that both AT2 and club cells decreased after irradiation (Figure 2d). The expression of 346 genes was altered in AT2 by irradiation, including 178 down-regulated and 168 up-regulated genes, many of which are mitochondrial genes (Figure 3a and Appendix A). The GO and KEGG pathway enrichment analyses showed that mitochondrial depolarization and oxidative stress response pathways were enriched in radiation-induced AT2 cells after RP (Figure 3b,c). Meanwhile, GSVA also indicates the enrichment of cGAS-STING signaling in AT2 cells of PR mice relative to the control (Figure 3d). Taken together, our results suggest that radiation treatment induced oxidative stress in lung tissue, and up-regulated numerous mitochondria genes involved in mitochondrial depolarization, which may activate the cGAS-STING signaling pathway.

We also found DNA damage leading to immune response may play an important role. As the pattern recognition receptor (PRR) related signaling pathways including Toll-like receptor NOD-like receptor signaling pathway and cytosolic DNA-sensing pathway were enriched (Figure 3b,c). Interestingly, the number of AT2 in the RP group decreased significantly, and many cell cycle regulatory genes such as CDKN1A, CCND1, and Ccng1 were up-regulated. The expression of Jun, JunD, JunB, and Rack1 decreased significantly (Figure 3a). These data suggest that despite some AT2 cells being eliminated by irradiation, the survived cells might be in a state of active proliferation to repopulate. Accordingly, ribosome and rRNA-related functions are also significantly enriched.

AT2 have been indicated to play an important role in pulmonary injury. In our study, this population also declined in the RP group (5.8% in RP lung tissue vs. 9.78% in control lung tissue; Figure 2b). Following subgroup analysis identified a total of 8 cell subtypes in AT2 (Figure 3d). Compared with the control sample, C2 was significantly enriched in irradiated tissue, while C1, C3, and C5 were significantly decreased. The C2 cells subtype mainly reflected the characteristics of radiation pneumonitis cells (Figure 3e,f). The differently expressed genes in C2 cells included those involved in cell cycle regulation and chromatin remodeling, such as Atrx, Ccnd1, and Zbtb20. The GO analysis of this subtype also showed an enrichment of cell adhesion-related pathways, as well as pathways responsive to oxidative stress, such as oxidoreductase activity, response to oxidative stress, and antioxidant activity (Appendix A). 

We next explored the AT2 states and cell transitions by inferring the state trajectories using Monocle2. This analysis showed that AT2 in the control lung tissue were located at the beginning of the trajectory path, whereas the irradiated AT2 were at a terminal state, revealing that AT2 underwent cell state transition in response to the stress induced by the irradiation (Figure 3f). 

Recently, Liu, et.al reported a new kind of bronchoalveolar stem cells (BASCs), located at bronchoalveolar duct junction (BADJ) and co-expressing the club cell markers and AT2 markers [37,38]. These markers were found specifically expressed in C1 (Appendix A), suggesting a high similarity with BASCs. In addition, the trajectory analysis showed that C1 predominated in the early stages, and cell state transitioned from C1 to C2 and C5 (Figure 3g–i). In general, the proportion of BASCs decreased significantly in irradiated groups, while the proportion of well-differentiated epithelial cells increased significantly.

### 3.3. Functional Changes of Fibroblasts and Vascular Endothelial Cells in RP

There were 349 fibroblasts identified from the RP and control lung tissue, accounting for 0.016% and0.017% of total cells in each group, respectively, (Appendix A). Gene expression profiling of the fibroblasts indicates the expression of 280 genes is decreased and 326 genes are elevated in the RP fibroblasts (Appendix A). The GO and KEGG enrichment analyses of these genes revealed that signaling of growth factor binding, fibril organization, focal adhesion, extracellular matrix (ECM)-receptor interaction, and Wnt signaling pathway, are enriched in RP fibroblasts. (Figure 4a,b and Appendix A). In addition, oxygen binding and hemoglobin are also significantly enriched, including hemoglobin complex and haptoglobin hemoglobin complex, which is consistent with the regulation of Hba-a2, Hbb-bt, Hba-a1, and Hbb-bs after RP. These may prompt RP to reduce the heme in response to ROS. Interestingly, cell proliferation terms were significantly enriched with 88 dysregulated genes, just as Cdkn1a was remarkably increased in the RP fibroblasts (Figure 4c and Appendix A). Together, these results indicate that fibroblasts in the RP lung tissue may enter a proliferative state and may reform the ECM, which could promote the occurrence of pulmonary fibrosis that usually happens following RILI.

The current opinion holds that endothelial cells (ECs) are the first responder to RILI^7^. In our study, we identified 288 vascular endothelial cells (VECs) in lung tissue samples, and the abundance of VECs was similar between the RP (184 cells, 1.36%) and control (104 cells, 0.014%) groups. DEGs of VECs showed that 476 genes were down-regulated, and 599 genes were up-regulated after irradiation (Appendix A). The GO and KEGG analyses of these genes showed enrichment of protein processing in the endoplasmic reticulum, TNF signaling pathway, and MAPK signaling pathway (Figure 4d,e), indicating that the RP VECs were active in protein processing, immune response, and cell proliferation. As in fibroblast, oxygen- and heme-related GO terms were enriched, and the oxidative phosphorylation signaling pathway participated in the RP response in VECs. Furthermore, the RP VECs also highly expressed Icam-1, an intercellular adhesion molecule and inflammatory mediator which belongs to the immunoglobulin superfamily (Figure 4f). Taken together, these results indicate that VECs play an important role in immune responses after irradiation.

### 3.4. scRNA-seq Identifies Five T Cell Subsets and Reveals an Altered T-Cell Subset Distribution in RP Mice

Our data showed that the proportions of T cells in lung tissue were slightly decreased in the RP group (Figure 1). These T cells could be further divided into five subtypes (Figure 5a). which were annotated as: CD4+ naive T cells (marker with CD4, Tcf7, Ccr7, Igfbp4, and Lef1), CD8+ naive T cells (Cd8a, Tcf7, Ccr7, Igfbp4, and Lef1), CD8+ effector T cells (Cd8a and Nkg7), Treg (Cd4, Foxp3, and Ctla4), proliferating T cells (Mki67, Nkg7, Gzma, Foxp3, and Ctla4), according to their gene expression characteristics (Figure 5b). The T cell subpopulations in PBMC and lung tissue were significantly different in that CD4+ naive T cells were predominant in both control and irradiated blood groups. Only CD4+ naive T cells, Treg, and CD8+ effector T cells were found in the blood, and the lung tissue showed a more diverse T cell population (Figure 5c). When comparing the composition of T cell subtypes in irradiated to the control lung tissue, the content of naive CD8+ T cells in the irradiated group was significantly decreased whereas the level of effector T cells was elevated more than three times, possibly due to the inflammatory state that induced functional activation of CD8+ T cells. 

Further GO pathway enrichment analysis in different T cell subtypes (Appendix A) showed that CD4+ naive T cells, CD8+ naive T cells, CD8+ effector T cells, and Treg were all enriched with T cell activation, suggesting that these T cell subtypes are all in activated states. CD4+ naive T cells, CD8+ naive T cells, and CD8+ effector T cells also highly expressed genes of ribosome-related pathways, indicating active protein synthesis, which may be related to the immune functions of these T cell subtypes. Meanwhile, Tregs highly expressed genes related to the regulation of cell–cell adhesion and T cell activation, which is consistent with their immune-regulatory functions.

Monocle2 analysis also revealed the dynamic immune states and cell transitions in T cells. We found the naïve T cells were mostly distributed throughout the early states in the trajectory path, whereas Treg cells and proliferative T cells were found mostly in terminal states (Figure 5d,e), demonstrating the transition of T cell states from naïve to activation.

We next exploited the active transcription factors and cis-regulatory elements in T cells from RP samples by SCENIC (single-cell regulatory network inference and clustering). In short, we identified 61 regions among T cells (Figure 5f). Some transcription factors were enriched in all T cell subtypes, including LEF1, JUNB, JUN, and FOS. In addition, TCF7 and FOXP1 were activated in naïve CD4+ T and CD8+ T cells. Finally, certain transcription factors demonstrated high transcriptional activity in specific T cells, such as EZH2 and YBX1 in proliferating T cells.

### 3.5. Specific Alveolar Macrophage Subpopulations Enriched in RP Mice 

Previous studies have shown that pulmonary macrophages play an important role in lung fibrosis and inflammatory response, which is usually a subsequent event of RP [39]. In the current study, we showed that the ratio of pulmonary macrophages raised slightly after irradiation (Figure 2c). The GSVA pathway enrichment analyses showed that the cGAS-STING pathway was activated in macrophages after IR (Figure 3d). Re-clustering of the 5173 macrophages revealed 11 different clusters (Figure 6a), each exhibiting a particular transcriptomic pattern (Figure 6b). 

A recent study suggests that expression of Fcn1, Spp1, and Fabp4 is enhanced in monocyte-derived, pro-fibrotic, and normal alveolar macrophage populations of idiopathic pulmonary fibrosis [40]. Interestingly, we found Fcn1^+^ macrophages were barely identified in either the control or irradiated group, suggesting that all the observed macrophages were not monocyte-derived, but tissue-resident alveolar macrophage populations. Among these Fcn1- macrophages, clusters C1 and C2, showed obvious expansion in the RP mice, both exhibited a Fabp4^low^ and Spp1^high^ feature. In contrast, the only Fabp4^high^ cluster C4 in the RP mice were nearly eliminated relative to the control mice. (Figure 6c–f). 

Moreover, there are also four Spp1^low^ Fabp4- clusters (C3, C5, C9, and C10) that highly expressed C1qc (Figure 6b,d,e), which is closely related to the regulation of immune cell functions [41]. These Spp1^low^, Fabp4^-^ C1qc^high^ clusters presented in samples from both control and RP mice, indicating a normal alveolar macrophage group that is not characterized by the recent classification criteria [32].

All the 11 macrophage clusters were subjected to Gene Ontology (GO) enrichment analysis (Appendix A) and Gene Set Variation Analysis (GSVA; Figure 7a) to further understand their biological functions. Notably, the GO analysis revealed that the clusters C1 and C2, which were enriched after irradiation, significantly enriched pathways related to lysosomes and the lysis function (Appendix A), indicating active phagocytosis. While GSVA showed that the C1 cluster highly expressed genes were frequently involved in ultraviolet (UV) response and DNA repair (Figure 7a), indicating that the activated phagocytosis function of the C1 macrophages might be reactive by irradiation. The GO analysis of the C2 cluster also indicated an enrichment of mitochondrial components and ATP metabolism (Appendix A). The cluster C4 that mainly presented in the control mice specifically expressed genes related to both the phagocytosis process and cell-substrate junction (Appendix A). A previous study shows that tight junction proteins expressed by airway-macrophage are essential for exchanging particles across the lung epithelium without injuring the airway wall, suggesting that C4 macrophages might play a fundamental role in normal lung functions [42]. 

We then employed SCENIC analysis to further investigate the gene regulatory mechanisms underlying the expression patterns of these macrophages. We found that activities of the following transcriptional factors in C1 cluster are specifically elevated: FOS, FOSB, JUN, JUNB, etc., which have been shown to play important roles in macrophage differentiation and activation [43,44]. Meanwhile, the C1 cluster also exhibited the activation of anti-inflammatory transcription factors EGR1 and NFIL3 (Figure 7b) [45,46], suggesting the anti-inflammation role of the C1 cluster upon irradiation. In addition, the SCENIC analysis also revealed the activation of alveolar-macrophage-specific transcription factors PPARG and CEBPB in C2 cluster enriching in the RP mice and the C4 cluster enriching in the control [40], and cluster C2 also showed enhanced activity of RUNX2 (Figure 7b). RUNX2 has been indicated to promote the role of macrophages in wound healing by regulating cholesterol and fatty acid biosynthetic processes [40], which is consistent with the enriched expression of fatty acid metabolism genes in C2 indicated by the GSVA analysis (Figure 7a). 

These 11 macrophage clusters were next subjected to RNA velocity analysis to understand their differentiation relationship. The results showed that different differentiation trajectories could initiate from different initial states. For example, the irradiation-specific clusters C1 and C2, and the control-specific cluster C4 all came from the same starting state C8 but bifurcated at the intermediate state cluster C11. Interestingly, as aforementioned Spp1^low^ Fabp4- C1qc^high^ clusters C3, C5, C9, and C10 represented a unique differentiation trajectory, which could start from either C5 or C9. It seemed that C5 was an intermediate state that could differentiate either into the RP-specific C2 or into the unique Spp1^low^ Fabp4- C1qc^high^ C3 and C10 (Figure 7c). Taken together, in the lung tissues of the irradiated mice, there was a loss of an Fcn1- Spp1- Fabp4^high^ normal alveolar macrophage population (C4), together with an expansion of two Fcn1- Fabp4^low^ Spp1^high^ macrophage populations (C1 and C2). Especially, C1 macrophages were likely an anti-inflammation population expanded in response to radiation, while C2 macrophages might play an important role in tissue wound healing after irradiation. Furthermore, we also found a distinct Spp1^low^ Fabp4- C1qc^high^ alveolar macrophage group, which may represent a distinct trajectory of mouse alveolar macrophage differentiation.

### 3.6. B cells in RP Mice Have Continuous Changes in Tissue

The subpopulation of B cells was also analyzed and a total of eight B cell subtypes were identified, including three known subtypes: plasma cells (C4), naive B cells (C6), and interferon-stimulated-genes (ISGs)-B cells (C8) (Figure 8a–c). It is noteworthy that plasma cells are mainly present in tissue samples but are absent in blood samples. Compared with control samples, the ratios of subgroups C1, C5, and C7 in the RP group were significantly higher (Figure 8d). Pseudotime trajectory analysis (Figure 8e,f) suggested that plasma cells were located at the terminal end of B cell development and differentiation, which is in line with the characteristics of B cell differentiation. Moreover, B cells of the pneumonia samples were distributed more continuously in different statuses than in control lung samples, which only located to a single state. This may be associated with a more diverse B cell subtype composition in the RP lung tissue, indicating that there are stronger relationships among the transition states of the cells from the pneumonia sample.

### 3.7. More Complex Cell–Cell Interaction in RP Mice

Finally, we sought to identify the potential cell crosstalk in RP lung tissue. The cell–cell interaction analysis revealed more complex intercellular communications in the RP group (Figure 9). We found that macrophages, mixed lymphocytes, NK cells, and neutrophils are dominant in the cell–cell communication network, which indicates the important roles of those cells in RP development. Comparatively, potential interactions among these cells are inactive in the control lung tissue. In addition, the fibroblasts could secrete MIF (migration inhibitory factor), GRN, CCL11, FAM3C, targeting CD74, TNFRSF1B, CCR2, FFAR2, and LAMP1 on macrophages, respectively. Among them, MIF is a multifunctional pro-inflammatory factor known to regulate innate immunity, promote the activation of macrophages, and form a positive feedback loop, by secreting TNF targeting monocytes, neutrophils, VECs, and AT2. 

## 4. Discussion

Radiotherapy is one of the major methods of local treatment for thoracic tumors, but the occurrence of RP limits the effective radiation dose. Studies have shown that the incidence of RP is 15.5–36% [47], which makes it a serious complication in the treatment of thoracic tumors. In recent years, there are numerous studies about the mechanisms of radiation-induced lung injury (RILI), each focusing on specific cell types, such as endothelial cells, T helper cells, as well as fibroblasts. However, the conventional methods are tissue-based, which cannot monitor the status of individual cell types in the context of the entire microenvironment.

Recently, scRNA-seq has been employed to analyze tissue samples with single-cell resolution. Cell types can then be annotated based on different expressions of characteristic marker genes in an objective and unbiased way. At the same time, the gene expression patterns of each individual cell type can be revealed in detail. In this study, we performed single-cell transcriptome analysis on PBMC and lung tissue from both irradiated and non-irradiated animals and obtained gene expression profiles of 43,062 single cells. Detailed bioinformatic analysis revealed cellular and molecular changes associated with RP. Our results showed that RP may be a relatively common and early injury due to activation of epithelial and stromal cells (such as ECs, AT2, Fib, etc.) and immune response events mediated by effector T cells and Fabp4^low^ and Spp1^high^ tissue-resident macrophages. To our knowledge, our study is the first attempt to use single-cell sequencing techniques to study acute radiation injury, which allows the identification of the key cells and molecules in the process of radiation damage in an unsupervised way.

At present, the recognized mechanism of RILI is the cytokine cascade theory in which radiation induces the production of reactive oxygen species (ROS) to damage lung parenchymal cells, including endothelial cells, macrophages, fibroblasts, AT2, etc., resulting in the secretion of a variety of cytokines [9]. Oxidative stress has long been considered one of the causes of acute and chronic radiation injury. As one product of irradiation, ROS can be generated in a variety of cell types as an important mediator in the occurrence of tissue injury and inflammatory environment, which may lead to sustained oxidative damage through the generation of active nitrogen species. For example, superoxide and nitric oxide were observed in the radiation-damaged alveolar cells, endothelial cells, and inflammatory cells to induce further prolonged local oxidative stress and sustained damage. Many attempts to reduce radiation damage focused on reducing oxidative stress via suppressing the production of ROS or with compounds to scavenger ROS after irradiation [48]. In fact, our results showed obvious ROS responses in those cells. Compared to the control, the level of oxidative stress response was enhanced in AT2 in the RP group, suggesting that AT2 in the RP group were under oxidative stress. Such as the oxygen binding, heme–copper terminal oxidase activity, and cytochrome-c oxidase activity were abnormal in AT2, and we also identified that the subpopulation of ROS responding AT2 were enriched in RP. In addition, our results showed the aberrant changes of these cells after radiation, including the loss of subpopulations that enriched in the normal lung as well as an alteration in transcriptomic patterns. Furthermore, the enrichment of the cGAS-STING signal pathway was detected in AT2 plasma cells of the RP group after radiation treatment. A previous study has suggested that the cGAS-STING signaling pathway is important in regulating inflammatory response and cancer progress. Additionally, the cGAS-STING pathway had been reported to be a promising therapeutic target in autoinflammatory, autoimmune, and degenerative diseases [49], indicating that inhibition of cGAS-STING might also be applicated in attenuating IR-induced lung injury. The cell-communication analysis showed complicated cell–cell interactions within the PR microenvironment.

Principally, the cytotoxic effect of radiotherapy relies on its DNA-damaging effect. After irradiation, damaged DNA or broken chromosomes are distributed in the cytoplasm and noted as micronuclei [50], triggering cytosolic DNA sensors, such as cGAS and AIM2, and the stimulation of DAMP-related immune responses [51,52,53,54]. The main role of the immune system is mediating the expression of pattern recognition receptors (PRRs) and the recognition of dangerous and invasive elements, which are distinguished in pathogen-associated molecular patterns (PAMPs) and damage-associated molecular patterns (DAMPs) [55]. It is known that DNA damage participates in radiation-induced inflammation, including liver injury and RILI, etc., via initiating innate immunity [56,57]. In this study, many downstream events of immune response were discovered in RP, but the expression of PRRs was kept at a normal level. This may be because radiation-induced PRRs events are very early responses of natural immunity, and our study is based on the eighth week after irradiation.

The loss of barrier function and vessel integrity always happens in RILI, induced by cell death, especially epithelial and endothelial cells, and leads to the reduction of micro-vessel density and oxygen perfusion [58]. It is generally believed that the pathogenesis of RILI is closely related to AT2. AT2 can secrete alveolar surfactant, which is important for maintaining the normal shape of the lungs and preventing alveolar collapse and the atelectasis effect [59]. There is evidence that the abundance of AT2 decreases after radiation, and mitochondria in AT2 are severely swollen [38]. Our study also showed that the proportion of AT2 significantly reduced after the irradiation. However, the remaining AT2 exhibited a proliferative feature, and no mitochondria function deficiency was detected at the transcriptomic level. In addition, there was an obvious reduction of stem AT2 in the RP lung tissue, while the proportion of highly differentiated AT2 increased. This hints that irradiation causes the death of stem AT2 and only the highly differentiated AT2 survive. Moreover, there is evidence that mitochondria in AT2 metabolize abnormally under oxidative stress, particularly in response to DNA damage and oxidative stress. Interestingly, our results indicated that IR treatment induced the oxidative stress response in lung tissue, up-regulated numerous mitochondria-related genes, and mitochondrial depolarization, which suggested that mitochondrial depolarization, initiated by oxidative stress, may activate the cGAS-STING signaling pathway. We found that expression of intercellular adhesion molecule 1 (Icam1), an important intercellular adhesion molecule and inflammatory mediator which belongs to the immunoglobulin superfamily and mainly exists in alveolar epithelium and capillary endothelium of lung tissue, was elevated in vascular endothelial cells (VECs) of RP lung tissue. The increase of Icam1 expression in lung tissue has also been observed in multiple previous studies. Hallahan et al. compared the response of wild-type mice and Icam1 knockout mice to chest radiotherapy, and found that the number of leukocytes in the lung of the knockout mice was significantly less than that of wild-type mice [60]. Fountain MD, et al. found that the expression of Icam1 in the lung tissue of normal rats was negative or weakly positive, but increased after irradiation [61]. Similarly, Van der Meeren A, et al. showed that the expression of Icam1 was significantly higher in the irradiated mice [62]. Sievert et al. found that Icam1 was continuously expressed in pulmonary microvascular endothelial cells for 20 weeks after local chest radiation [63]. In general, radiation can induce the production of Icam1 by ECs and chemotactic leukocyte adhesion, and cause an inflammatory reaction.

Previous studies have shown the role and potential of clinical application of Th1/Th2 imbalance in RP [64]. However, the role of killer T cells and regulatory T cells in RP has not been clarified clearly. We observed a significant expansion of CD8+ effector T cells, slightly expanded regulatory T cells and proliferating T cells, and a drastic decrease of naive CD8+ T cells after irradiation in the current study. Proliferating T cells often increase in pulmonary fibrosis, asthma, and RP [64]. These suggest that T cells may respond to the irradiation via a transformation of subtype composition. Therefore, we hypothesized that activation of tissue-resident CD8+ T cells plays a crucial role in RP, and that also can induce the recruitment of T cell populations from the blood.

In acute inflammation, macrophages play a variety of functions, including the production of inflammatory cytokines and repair molecules, and phagocytosis of cellular debris. Therefore, macrophages are attractive targets for therapeutic interventions in inflammatory diseases. Several previous studies have demonstrated that tissue-resident macrophages proliferate in response to acute inflammation [65]. Interestingly, in our study, all the macrophages found in both control and RP lung tissues were identified as tissue-resident macrophages. Moreover, the enrichment of the cGAS-STING signal pathway in macrophages of the RP group after irradiation indicates its fundamental role in RP pathogenesis. It is reported that kidney fibrosis, induced by inflammation, was closely related to mitochondrial damage and the activation of the cGAS-STING pathway [66]. Hence, the activation of the cGAS-TING pathway in macrophages is probably related to lung fibrosis and the inflammatory response of RP mice. These macrophages expanded in RP lung tissue, which echoed previous studies. 

Furthermore, we also observed different representative macrophage subpopulations in control and RP lung tissue. A Fabp4^high^ macrophage cluster that likely played a fundamental role in normal lung functions enriched in the control lung tissue but largely decreased in the RP lungs, which may be associated with lung injuries caused by irradiation. In the RP lung tissue, however, we observed enrichment of two specific macrophage clusters, one of which was likely involved in the ultraviolet (UV) response, DNA repair, and anti-inflammation functions, while the other may be essential in tissue wound healing. These indicate that specific macrophage subpopulations not only proliferate in response to the irradiation but probably also contribute to tissue repair after the irradiation.

Our single-cell sequencing study reveals the landscape in the lung and PBMC with RP and provides an insightful framework for understanding the cellular and molecular mechanisms and transcriptomic patterns of RP. It is the first single-cell-level analysis to understand the role of complex interactions between multiple types of immune cells in the context of RP, which could serve as a basis for probing cellular interactions underlying the immune responses during the development of RP. We believe that such detailed information could provide novel insights into the pathogenesis of RP and for the development of advanced therapies for better disease control. However, more validation assays need to be carried out to verify the conclusions in future studies.

## 5. Conclusions

Our results provide the first single-cell profiling of RP in mice, offering valuable insights into the cellular and molecular mechanisms of the development of RILI.

## Figures and Tables

**Figure 1 antioxidants-11-01457-f001:**
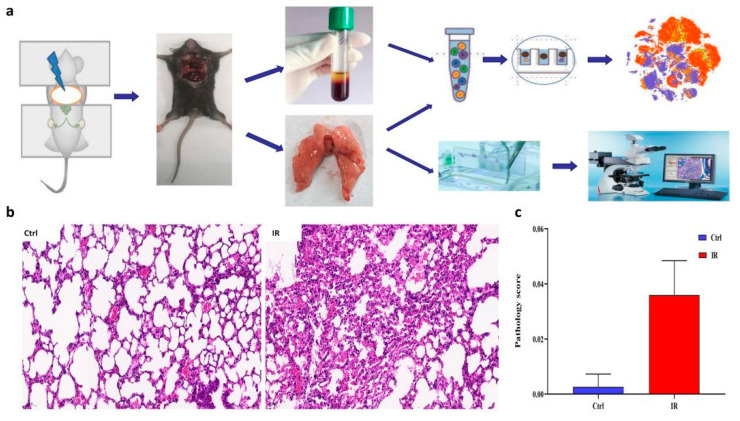
Overview of the lung tissue of radiation pneumonitis (RP) mice and control mice. (**a**) Schematic diagram of experiment design. (**b**) H&E staining of representative sections is shown. (**c**) A summary of histology scores assigned to each group.

**Figure 2 antioxidants-11-01457-f002:**
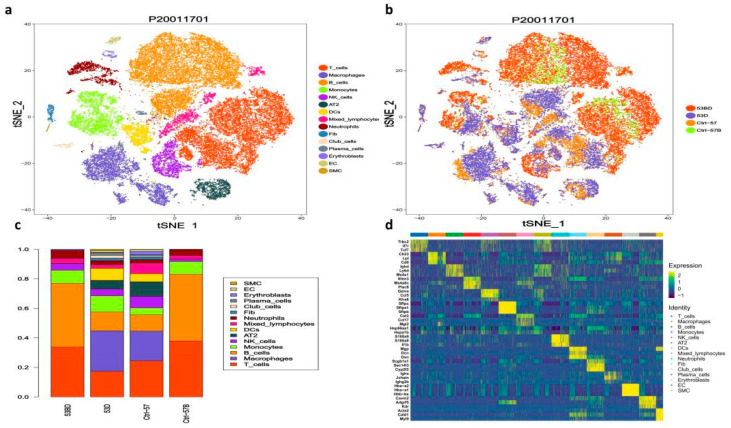
Single-cell transcriptional profiling of the lung tissue and peripheral blood mononuclear cells (PBMC) of radiation pneumonitis (RP) mice and control mice. (**a**) T-SNE for all cell types, including distinct lymphocytes, myeloid cells, type 2 alveolar cells (AT2), fibroblasts, smooth muscle cells, and vascular endothelial cells (VECs). Cell types and samples are indicated on the right. (**b**) The same T-SNE as in (**a**), colored by samples, including (CN, the lung tissue sample from the control mouse); CN-blood, the PBMC sample from the control mouse; RP, the lung tissue samples from the irradiated mice; RP-blood, the PBMC samples from the irradiated mice; (**c**) relative proportion of each cell cluster across 4 samples as indicated. (**d**) Heatmap showing the differentially expressed genes of different clusters, genes, and cells ordered by hierarchical clustering.

**Figure 3 antioxidants-11-01457-f003:**
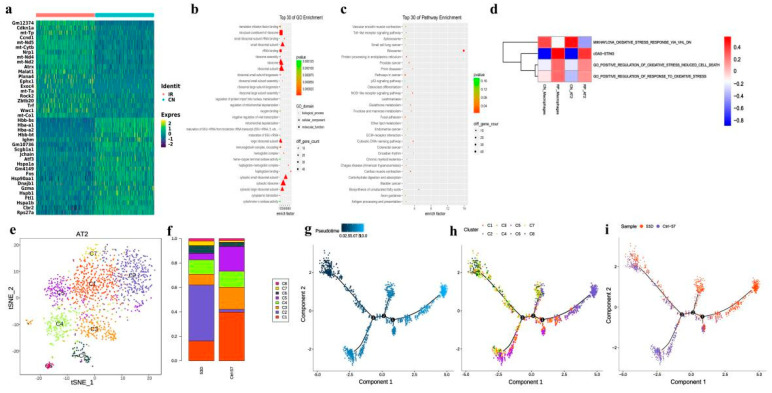
Brief description of 1811 epithelial cells and simulation of the development trajectory of epithelial cells. (**a**) Heatmap showing scaled expression of differentially expressed genes defining the epithelial cells of PR and CN. (**b**) Dotplot showing the representative top 30 GO terms enriched in AT2 cells. (**c**) Dotplot showing the representative top 30 pathway terms of KEGG enriched in AT2 cells. (**d**) The heatmap demonstrates the enrichment of top 4 pathway activities in AT2 cells via GSVA of PR mice relative to the control. (**e**) The t-distributed stochastic neighbor embedding (t-SNE) plot of the 8 types of epithelial cells. (**f**) Relative proportion of each cell cluster across PR and CN as indicated. (**g**–**i**) Unsupervised transcriptional trajectory of AT2 subcluster cells from Monocle (version 2), colored by pseudotime, cell subsets, and samples.

**Figure 4 antioxidants-11-01457-f004:**
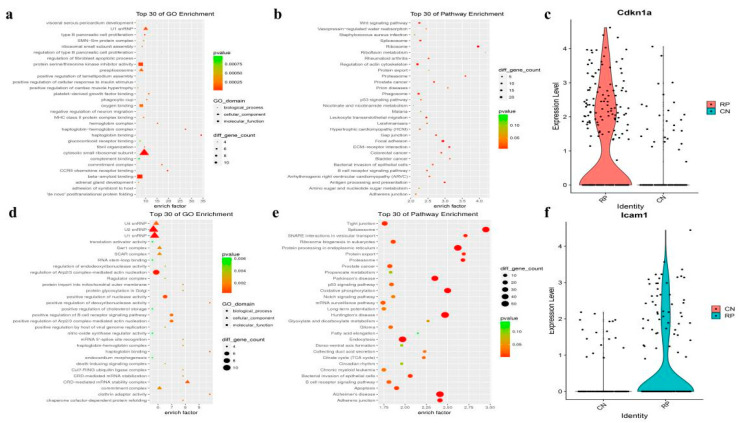
The fibroblasts and vascular endothelial cell markers and pathway analysis. (**a**,**b**) The GO analysis of up-regulated DEGs shows top30 highlighted pathways in RP fibroblasts cells (*n* = 4, hypergeometric test, adjusted *p*-values obtained by Benjamini-Hochberg procedure). (**c**) Violin plots showing the expression level of representative novel identified markers across the fibroblasts cell types. (**d**,**e**) Dotplot showing the representative top30 pathway terms of KEGG enriched in vascular endothelial cells (VECs). (**f**) Violin plots showing the expression level of representative novel identified markers Icam1 across the endothelial cells.

**Figure 5 antioxidants-11-01457-f005:**
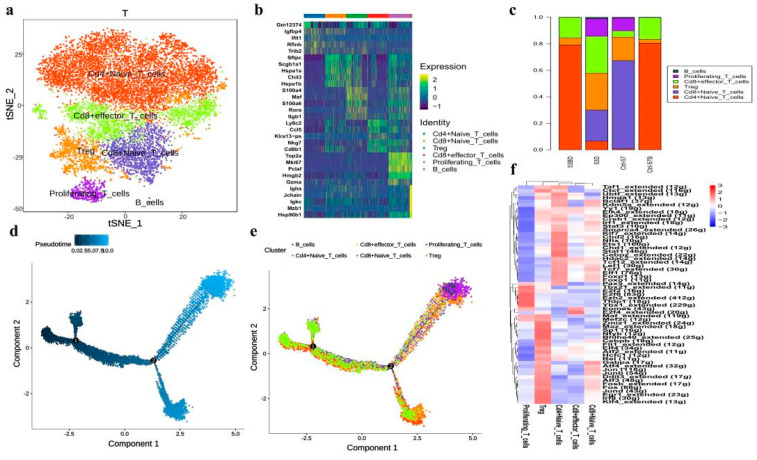
Single-cell transcriptional profiling of T cells and trajectory branch. (**a**) The T-SNE presentation of the T cell clusters and B cells. (**b**) The heatmaps of hierarchically clustered top 10 differentially expressed genes (DEGs) across five groups of T cells and one group of B cells. The gene names were listed to the right. (**c**) The bar plot shows the percentages of T cell clusters and B cells in each studied subject. (**d**) Simulation of the differentiation trajectory of T cells subtypes. (**e**) Cell source transition. (**f**) The heatmap exhibits the average regulon activities of transcription factors in different T cellular types, the color key from blue to red indicates relative expression levels from low to high.

**Figure 6 antioxidants-11-01457-f006:**
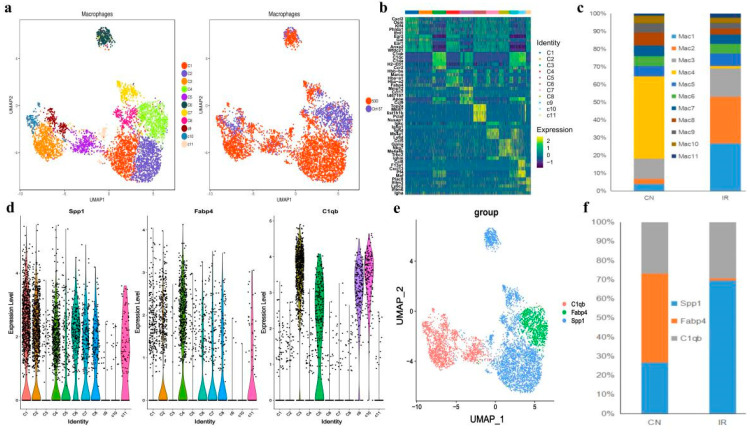
Single-cell transcriptional profiling of macrophages cells. (**a**) The t-distributed stochastic neighbor embedding (t-SNE) plot of the 11 identified cell types in macrophages cells. (**b**) Heatmap of the DEGs among the 11 cell subtypes, where the colors from red to blue represented alterations from high expression to low expression. (**c**) The fraction of cells for macrophages subsets percentage of subsets across individual donors. (**d**) Violin plots showing the normalized expression levels of 3 representative canonical marker genes across the 11 clusters. (**e**) t-SNE of single-cell profile with each cell color-coded by representative novel identified markers of C1qc, Icam1, and Fabp4 across the macrophage populations. (**f**) The bar plot shows the relative contributions of each cluster as in E between PR and CN.

**Figure 7 antioxidants-11-01457-f007:**
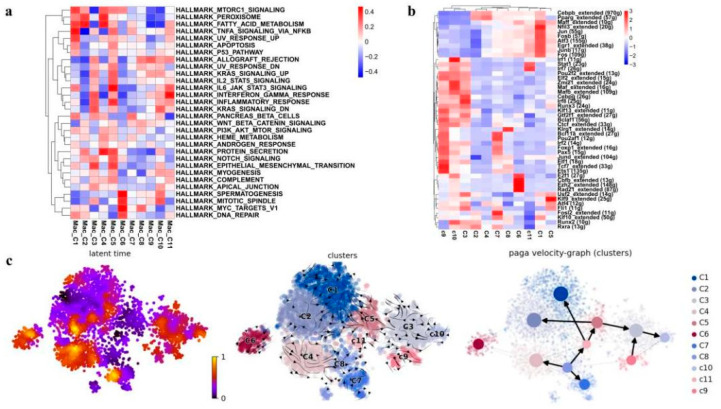
Identification and characterization of macrophage cell populations. (**a**) Heatmap showing the representative pathway terms of hallmark enriched in macrophage cellular subgroup. (**b**) The heat map exhibits the average regulon activities of transcription factors in macrophage cellular subgroup (top 10 regulons in each cluster were selected). The number in the parenthesis shows the downstream genes regulated by the corresponding transcription factors. (**c**) RNA velocity analysis of macrophage cellular subgroup with velocity field projected onto the PAGA plot of macrophages subclusters (**right** panel). Arrows show the local average velocity evaluated on a regular grid and indicate the extrapolated future states of cells. Left panel: RNA velocity pseudotime analysis revealing the latent time of macrophages subclusters.

**Figure 8 antioxidants-11-01457-f008:**
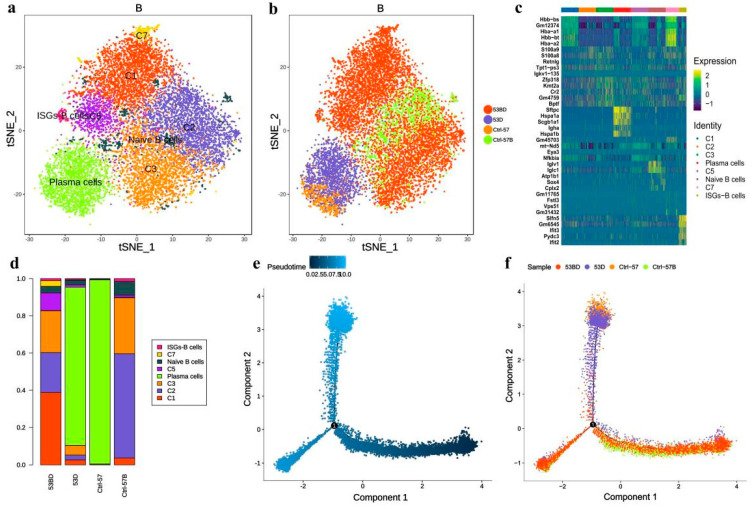
Single-cell transcriptional profiling of B cells. (**a**,**b**) t-SNE of B cells profile with each cell color-coded for associated cell types and sample type. (**c**) The heatmaps of hierarchically clustered top 10 differentially expressed genes (DEGs) across 8 groups of B cells. The gene names were listed to the right. (**d**) The fraction of B cells subtypes for each sample. Simulation of the development trajectory of B cells and the analysis of gene expression pattern inferred by Monocle2, (**e**) pseudo-trajectory of B cells, (**f**) cell source transition.

**Figure 9 antioxidants-11-01457-f009:**
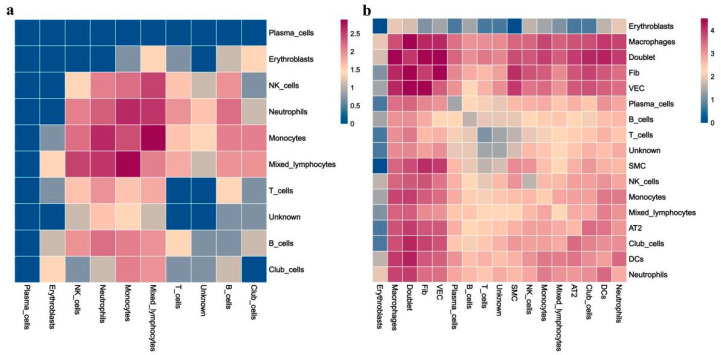
Heatmaps of identified potential cell–cell crosstalk in control (CN) (**a**) and RP groups (**b**).

## Data Availability

Data is contained within the article or Appendix A. For more details, please check: https://www.mdpi.com/journal/antioxidants/instructions#suppmaterials (accessed on 20 July 2022).

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
