# Peer review of "Single-Cell Transcriptome Analysis of Radiation Pneumonitis Mice"

_antioxidants, 2022, doi:10.3390/antiox11081457_

Round 1

Reviewer 1 Report

1. The title should in include the phrase "lungs" or whatever it is that was sequenced.

2. References to the mouse model should be given in the methods section.

3. Fig 1B is unclear and appears to be distorted.

4. Almost all of the figures are too small to read.

Reviewer 2 Report

The Authors responded successfully to all of my comments, therefore I would suggest accepting the manuscript for publication in Antioxidants. However, although I stated that I do not feel qualified to judge the English language, I think that extensive editing of language and style is needed. Therefore, I would say "accept after minor revisions" (Language and style). I would leave this decision to the Editors, and I do not need to check the manuscript anymore! 

Reviewer 3 Report

The authors attempted to use blood and lung scRNAseq to reveal transcriptomic landscapes associated with RP.  Overall data analysis methods are good and done with expected rigors.

The idea is novel and can inform generally topics related to radiation-induced lung injury.

The study is limited because one 1 mouse in the control and 2 mice in the RP group are used.  Fibrosis is highlighted as one of the outcomes but the degree of fibrosis in the lung tissue is not assessed in this particular animals (even though a reference was added instead).  

Overall, RP seems to increase CD45+ leukocytes in the lung, and this will squeeze out CD45- cells in a normalized mixture of cells (7000 or so per sample).  The significant downstream bias from this method is not considered, and particularly observations noted in rare populations (AT2, fibroblast, some B cell subpopulation) are concerning for making incorrect conclusions due to this type of bias.

While I think I understand why, the role of peripheral scRNAseq seems at best nebulous and confusing.  And the authors are not really making explanations or attempting to connect what is in peripheral blood vs. lung, limiting usage to a pure description of observations.  Either use the data more extensively to add to the conclusion or remove the blood scRNAseq data.

Line 49: Prob more clear statement is "occurs 1 year after treatment?"

Line 133: Which gradient was used for scRNAseq with peripheral blood? A certain cell population was enriched?  Does the gradient skew the cell composition to particular cell types?  Need more details and validation of what cells are eventually imputed for scRNA sequencing.

Fig 1B seems not proportional in the natural ratio (height and width: width is stretched... needs correction): not publishable quality

Line 232: Supplemental figure 1A-1C does not have cluster number but cell type description. Add cluster number to this.

Line 272-278-Data in Fig 1 is difficult to analyze: If XRT causes inflammation, changes in blood will be directly reflected in changes in leukocyte composition, but if XRT causes inflammation in lung tissue, leukocytes (CD45+) will go up, thus, the proportion of CD45- cells will go down. For example, if we presume that in control mice, 60% of 7000 lung cells will be CD45+, 40% of 7000 cells will be CD45-, then in XRT exposed mouse lung, inflammation will increase the composition of CD45+ cells to (let's presume to 80%) and this will reduce the composition of the CD45- cells to 20%.  Since each sample is normalized by the same total number of cells, this will confound by decreasing the CD45- cell population.  Then how do we know the accuracy and actual magnitude of the reduction in non-inflammation cells?  Either CD45- (maybe Epcam+) cells need to be enriched to see if AT2 cells are indeed reduced in the RP group?  Also will need a flow of the whole lung to quantify the actual number of AT2 cells.  Because there is a concern that the representation by AT2 may be skewed (losing gene signature in the current method), the DEG results are concerning for potential bias and confounding factors.

Line 280: again the 5.8% in RP and 9.78% in CTL lungs may be biased numbers due to the composition of the cell count that did not consider over-enriched leukocytes in RP lungs.  The conclusion in lines 298-302 again falls into this bias.

Line 329-331: how is the %composition of VEC similar? 184 vs. 104 cells (when the total cells sequenced are about the same around 7000?)  Is the math correct?

Line 358: Complete disappearance of CD8+ naive cells noted in RP group.  Interesting but with this kind of change, I need to see some validation of this transcriptomics being biologically real.. meaning, flow corroboration.

I think the authors are trying to suggest specific lymphocyte trafficking from circulation to lung by showing blood and lung together.  More rationale and explanation are desired, and if to be highlighted as a reason for skewing T cell population is related to this correlation, secondary validation is required (flow).

Lines 391-407: authors highlighted Fabp4 and Spp1 as featured genes to determine resident vs. recruited macs.  I think more classical Mac conditioning transcription factors will need to be considered such as pparg, lxra, lxrb, maf, mafb, mef2a, mef2b, mef2c, mef2d, and some gata genes.  If particularly pparg positive macs are there, there must be some connection to Ccr2+Ly6c+ monocytes being recruited to the lung and being conditioned as lung mac?  The analysis here seems a bit too superficial as the authors immediately start to focus more directly on a subset of mac...

Line 472-474: highlighted B cell clusters C1, C5, and C7 to be over-represented in RP group.  Fig 7D is over-represented by plasma cells that these three clusters are almost indistinguishable at first and also unclear if they can be even biologically significant. How many cells are we talking about here? If really small in number, need to determine if these cells can be biologically relevant by at least with flow. Especially based on Fig 7A t-SNE plot, C7 may be a part of C1?  C1 vs. C7 top 10 markers seem similar in Fig 7C except for the difference in their intensity?  More validation is desired to confirm that these two clusters are indeed different?

While appreciating the author's proposed mechanistic diagram (Fig 9), without any validation of even the cell types of corroborating methods (like flow), this can be misleading and incorrectly concluded.  Therefore, I would delete this figure.

Line 599-606 will need to be either restated or deleted after further validation of dramatic changes in T cell populations is completed.  CD4 marker is particularly promiscuous in its transcriptomic expression pattern, thus, more validation is needed before making any conclusion on this sub-topic.

Line 614-619: Since the authors are highlighting cGas-Sting path to connect mitochondrial damage and activation leading to fibrosis and inflammation, more corroborating studies are requested or reduce the enthusiasm on this conclusion.

Reviewer 4 Report

Yang and colleagues present an analysis on single-cell transcriptome based on 43000 cells in order to describe the molecular landscape of radiation pneumonitis in a mouse model. This soundly written paper covers a topic of utmost interest to the scientific community engaged in thoracic oncology. Therefore, I recommend it for publication after considering the following issues below:

cGAS-STING pathway

·         Is there any clinical evidence that the cGAS-STING pathway is activated in patients with RP?

·         How does this pathway interact – if at all – with known molecular markers of RP/RILI, such as TGF-b or IL-6 and others that belong to the ROS cascade?

·         Line 550: The authors mention that radiotherapy results in DNA damage. In how far does this pathway interact with DNA damage sensors and repair enzymes?

·         Could you please clearly differentiate between acute RP and late radiation fibrosis as the results of this study are based on a situation 8 weeks after irradiation, which is – clinically speaking – acute RP.

Minor:

Line 1: Radiotherapy is not an “auxiliary therapy”, please use the term “adjuvant” throughout the manuscript

Line 503: Please quote literature with the numbers 15.5% - 36% mentioned in this line.

Line 504: “they’re” should be “they are”

Line 522: Please quote literature that refers to the ROS cytokine cascade.

Reviewer 5 Report

Specific Comments:

Abstract:

The abstract should cite the genes that are unique to specific cell types, rather than give an overview.

Introduction:

There needs to be a discussion between acute radiation pneumonitis and late fibrosis.

Materials and Methods:

This section is well written.  The description of how the single-cell transcriptome is analyzed is good.

Results:

Figure 1 is too busy. It should be subdivided into multiple figures. The description of AT2 and Club cells needs to be amplified. 

The paragraph, which starts with the sentence “We also found DNA damage leading to immune response may play an important role since the pattern recognition receptor (PRR) related signaling pathways including Toll-like receptor signaling pathway, NOD-like receptor signaling pathway and cytosolic DNA-sensing pathway were enriched (Fig. 2B-C).” 

This is a run-on sentence and combines elements that should be in the discussion with multiple sentences that should be in the results.  This is lines 259-276.  The quality of the grammatic English is poor.

Figure 2 shows transcription profiling of epithelial cells. This figure is too busy and should be subdivided.

The same holds for Figure 3 with fibroblasts and endothelial cells.

The same holds for Figure 4. The figures are much too dense and too busy.

The grammatic English describing the results in figures should be modified.

Figure 5 is also too busy.

The same holds for Figure 6.  There is so much raw data in these figures that it is difficult for a reader, let alone a reviewer to analyze the results. 

The data in the heatmap in Figure 8 is not clear.

Discussion:

The discussion begins with a description of radiotherapy.  Since the article regards mice and radiation pneumonitis. It is not clear how this section relates to the manuscript.

Figure 9 is not related to the manuscript and is a theoretical consideration and more appropriate for a review article.

Round 2

Reviewer 1 Report

None

Reviewer 2 Report

I have no further comments. I think that manuscript should be published.

Reviewer 3 Report

The authors revised the manuscript within their ability.    

Reviewer 4 Report

I have nothing to add.

Reviewer 5 Report

Manuscript is now acceptable for publication.